# Early Diagnosis of Central Disorders Mimicking Horizontal Canal Cupulolithiasis

**DOI:** 10.3390/brainsci13040562

**Published:** 2023-03-27

**Authors:** Paula Peña Navarro, Sofía Pacheco López, Cristina Nicole Almeida Ayerve, Susana Marcos Alonso, José Manuel Serradilla López, Santiago Santa Cruz Ruiz, José Carlos Gómez Sánchez, Diego Kaski, Ángel Batuecas Caletrío

**Affiliations:** 1Neurotology Unit, ENT Department, University Hospital of Salamanca, IBSAL, 37007 Salamanca, Spain; 2Neurotology Unit, Neurology Department, University Hospital of Salamanca, IBSAL, 37007 Salamanca, Spain; 3Department of Clinical and Movement Neurosciences, Institute of Neurology, UCL, London WC1N 3BG, UK; 4ENT Department, Faculty of Medicine, University of Salamanca, 37007 Salamanca, Spain

**Keywords:** benign paroxysmal positional vertigo, horizontal semicircular canal, central positional nystagmus, differential diagnosis

## Abstract

Background: Horizontal Canal Cupulolithiasis (hc-BPPV-cu) can mimic a pathology of central origin, so a careful examination is essential to prevent misdiagnosis. Methods: Retrospective cross-sectional cohort study of 45 patients suffering from suspected hc-BPPV-cu. We recorded whether patients first presented through an ENT Emergency Department (ED) or through an Outpatient Otolaryngology Clinic (OC). Results: We found statistically significant differences (*p* < 0.05) between the OC versus the ED in relation to the time between symptom onset and first assessment (79.7 vs. 3.6 days, respectively), the number of therapeutic maneuvers (one maneuver in 62.5% vs. 75.9%, and more than one in 25.1% vs. 13.7%), and multi-canal BPPV rate (43.8% vs. 3.4%). hc-BPPV-cu did not resolve in 2 patients (12.5%) from the OC and in 3 (10.3%) from de ED, all of which showed central pathology. Discussion: There are no prior studies that analyze the approach to hc-BPPV-cu in the ED. The benefits of early specialist input are early identification of central positional nystagmus, a decrease in symptom duration, reduced number of therapeutic maneuvers required for symptom resolution, and lower rates of iatrogenic multi-canal BPPV. Conclusion: A comprehensive approach to hc-BPPV-cu in the ED allows both more effective treatment and early identification of central disorder mimics.

## 1. Introduction

Benign Paroxysmal Positional Vertigo (BPPV) is a type of vertigo of peripheral origin that is generally triggered following a change in head position. It is produced by abnormal mechanical stimulation of vestibular receptors due to detachment and displacement of the aberrant otoconia within the semicircular canals, in the inner ear, giving characteristic symptoms of brief positional rotational vertigo and instability [1].

BPPV is the most common peripheral vestibular condition, accounting for approximately 25% of all peripheral vertigo, with an estimated lifetime prevalence at 2.4% in the general population, and a 1-year incidence at 0.6% [2]. It more frequently affects women in the sixth decade of life, and its most frequent etiology is idiopathic, followed by others such as head trauma, migraine, Ménière’s disease or vestibular neuritis [3,4].

BPPV most frequently affects the posterior semicircular canals (pc-BPPV) (representing 60–80% of BPPV cases), followed by horizontal canal BPPV (hc-BPPV) which has a prevalence of 26% [5] (it can reach up to the 40% of cases, especially in elderly patients), and lastly the anterior canals (ac-BPPV) with a prevalence of 1–2% [6]. In BPPV, free-floating otoconia within the semicircular canals give rise to the canalolithiasis variant, whereas otoconia adherent to the cupula is responsible for cupulolithiasis, a rarer form. As for the other canals, horizontal Semicircular Canal Cupulolithiasis type BPPV (hc-BPPV-cu) is less frequent than canalolithiasis and can be diagnostically challenging.

Care protocols have been developed to increase the diagnostic yield of BPPV in the emergency department [7,8]. Thus, in a patient with typical symptoms of rotational vertigo, vegetative symptoms, and unsteadiness, nystagmus provocation maneuvers are performed, with careful attention to the characteristics of the nystagmus. Where they are diagnostic of BPPV, particle repositioning maneuvers are carried out, clearly identified as the most effective treatment for this pathology [9,10]. It is widely established that pc-BPPV is diagnosed with the Dix-Hallpike maneuver, and treated using the Epley maneuver [11], and that in ac-BPPV, head hyperextension and Hain-Yacovino manoeuvres are used [12], respectively [13]. The diagnostic maneuver in hc-BPPV is the McClure-Pagnini test or Supine Roll Test, which carries the highest accuracy of lateralization [14,15], with the addition of the Bow and Lean test in hc-BPPV-cu. However, some controversy exists regarding the optimal treatment maneuver for hc-BPPV-cu since there have been several attempts over the years to describe the most effective one; from the first ones described as prolonged forced position in 1994 [16], through more current ones such as the Gufoni maneuver documented for the first time in 1998 [17] (probably the most widespread, with proven immediate symptomatic recovery [18]) and the most recently described by Zuma e Maia, with its latest modification published in 2020 [19].

Thus, hc- BPPV-cu is less frequent and can often be difficult to correctly define clinically, despite the several attempts to characterize the ocular motor signs using a range of different manoeuvres.

More pressingly, hc-BPPV-cu may induce spontaneous (non-positional) nystagmus [20,21] and has been shown to mimic central disorders [22,23]. However, the differential diagnosis of hc-BPPV-cu in the context of these findings remains a clinical challenge, despite increasing recognition of the importance of differentiating between this benign peripheral syndrome and central disorders.

Positional nystagmus is defined as the nystagmus generated by a change in head position with respect to gravity [24]. This type of nystagmus may be triggered by peripheral vestibular pathology (more frequently in relation to BPPV) [25], or by central pathology, known as central positional nystagmus.

Therefore, the aim of this study was to evaluate the clinical differences between hc-BPPV-cu versus a disorder of central origin with similar clinical presentation, across Emergency Department (ED) and Outpatient Otolaryngology Clinic (OC) settings.

## 2. Materials and Methods

This was a retrospective cohort study. We compared two patient cohorts that presented with BPPV to either the Outpatient Otolaryngology Clinics (OC; Group A) or the ENT Emergency Department (ED; Group B) of the Otorhinolaryngology Service of Salamanca University Hospital, with regard to diagnostic workup and therapeutic outcomes. Adult patients presenting between January 2010 and December 2021 with positional vertigo and horizontal nystagmus on positioning were included. A clinical diagnosis of hc-BPPV-cu was confirmed following detailed review of the clinical notes, and adherence to the diagnostic criteria as established by Bárány Society [24]. All patients were followed up with diagnostic and therapeutic maneuvers until full resolution of positional symptoms or until a different diagnosis for the positional symptoms was reached. In the absence of additional red flag symptoms (central nervous system signs or symptoms, atypical nystagmus, or vomiting) [13], a Magnetic Resonance Imaging (MRI) scan was routinely performed if positional symptoms persisted after a 5th attempted therapeutic maneuvers [9,13,24]. Patients who developed hc-BPPV-cu in the context of a multi-canal BPPV (mc-BPPV) were also included. Patients with incomplete clinical data or follow-up were excluded.

The recruiting ED follows our own protocol for “Patient Care with Benign Paroxysmal Positional Vertigo”, in which a patient with symptoms compatible with this disorder is assessed by the on-call Otolaryngologist within 4 h, undergoes diagnostic and therapeutic positional maneuvers and is followed-up every 10 days with performance of a new repositioning maneuver in case of persistence of hc-BPPV-cu (assessed with the diagnostic maneuver), until the complete resolution of the condition. Patients evaluated in OC were reviewed by the same group of otolaryngologists following a similar evaluation protocol as in the ED and follow up. However, patients in this group underwent additional routine 3-monthly follow-up for a minimum of 1 year given possible rates of BPPV recurrence. In both groups, hc-BPPV-cu was diagnosed using the McClure maneuver with observation of apogeotropic horizontal nystagmus, and on occasions when the laterality of the affected canal was in doubt, the Bow and Lean test was added, thus indicating the direction of the nystagmus obtained in the extension of the affected side. In both groups, hc-BPPV-cu treatment was primarily done using the Gufoni maneuver, and the Zuma e Maia maneuver in the last years, although during the first period of this study the prolonged forced position was also used.

Data were collected from each patient regarding their age, sex, history of BPPV, origin as to whether the patients access the Otolaryngologist through ED or OC, and clinical features of the hc-BPPV-cu regarding its etiology, diagnosis, complementary tests (MRI), treatment and evolution. Other variables collected included BPPV side, past medical history, personal history of vestibular disorders (including BPPV), trigger (where known), associated otological symptoms, pharmacotherapy and residual symptoms. Data collection was carried out using Microsoft Excel 2016, corresponding to the Office package of Microsoft Corporation, version 18.2110.13110.0.

Statistical analysis was performed using the IBM SPSS Statistics version 20 statistical software platform. Initially, descriptive statistics were implemented based on frequency tables of each variable described, with their percentage, mean and standard deviation, and median. Each variable within the cohort (OC or ED) was compared using contingency tables, evaluating their concordance. Subsequently, the statistical significance of the comparison of each variable in each cohort was evaluated with parametric tests, using a significance *p*-value < 0.05.

## 3. Results

This study included a total of 45 patients with a definitive diagnosis of hc-BPPV-cu who met the established inclusion criteria. Main demographic and clinic details are presented in Table 1.

Other variables were analysed, such as BPPV side, personal history, vestibular disorder history, etiology or trigger (if known), associated otological symptoms, pharmacotherapy and residual symptoms. The analysis of these variables did not offer relevant results in this study.

### 3.1. Outpatient Clinic

Cohort A (OC) included 16 patients, 62.5% female, with a mean age of 70.5 years (SD 12.65), and an average of 79.7 days between symptom onset and first evaluation. 31.3% of the patients in this cohort had a prior history of BPPV, affecting the posterior semicircular canal in all cases. In one patient of this subgroup, there was a combination of pc-BPPV and hc-BPPV-cu. 43.8% of patients in the Cohort A had multicanal BPPV (mc-BPPV) at the time of hc-BPPV-cu diagnosis, involving a combination of posterior canal and horizontal canal. In 62.5% of cases, hc-BPPV-cu resolved with a single maneuver (the first particle displacement maneuver performed after diagnosis), with the Gufoni maneuver being the most used (37.5%). In contrast, 25.1% of the patients required more than one treatment maneuver for resolution, with 25% of them requiring a change of maneuver, and in 2 patients (12.5%) the hc-BPPV-cu persisted for the duration of follow-up, both of whom had a central cause identified on further brain imaging, shown later (Figure 1).

### 3.2. Emergency Department

In Cohort B, 29 patients (64.4% of the total) were seen in the ENT ED, according to the previously described protocol. Of these, 44.8% were female, with a mean age of 67.07 years (SD 14.09) and a mean of 3.6 days of days from symptom onset to evaluation. 27.6% of the patients in the cohort presented a prior clinical history of BPPV (6.9% pc-BPPV and 10.3% hc-BPPV). One patient (3.4%) in Cohort B had mc-BPPV at the time of hc-BPPV-cu diagnosis, involving a combination of posterior canal and horizontal canal. In 75.9% of the cases treated in the ED, the hc-BPPV-cu was resolved with the first maneuver performed, the majority being Gufoni’s (69% of the cases). We found that 13.7% of the patients in the cohort required more than one maneuver to resolve their pathology, a change of maneuver being necessary in 10.3% of these at the next follow-up. In 3 patients in the cohort (10.3% of the total patients in Cohort B) the hc-BPPV-cu was not resolved despite repeated maneuvers, all of whom had a central cause identified on subsequent imaging (Figure 2)

In this way, a statistically significant difference (*p* < 0.05) was observed between ED and OC cohorts for the time between symptom onset and first evaluation, the number of maneuvers necessary for the resolution of hc-BPPV-cu and the existence of a mc-BPPV at the diagnosis of hc-BPPV-cu (Figure 3).

### 3.3. Central Disorders

Notably, hc-BPPV-cu was not resolved in 2 patients attending the OC and in 3 from the ED. All of these showed central pathology on subsequent MRI brain scans, thus mimicking hc-BPPV-cu, but presented with additional neurological symptoms. In the OC cohort, one patient with a hemipontine stroke (Figure 1) had associated cardiovascular risk factors (arterial hypertension and dyslipidemia) together with right-sided hearing loss and marked unsteadiness. The second patient with cerebellar atrophy, associated instability and a cerebellar pattern videonystagmogram and a left vestibular deficit of 28%. In the ED cohort, one patient had a stroke due to bilateral Posterior Inferior Cerebellar Artery (PICA), associated with cardiovascular risk factors (arterial hypertension, atherosclerosis and diabetes mellitus), unsteadiness and gait ataxia; another patient had dissection of the vertebral artery, associated with severe neck pain; the third patient was found to have metastasis in the cerebellar hemisphere, in the context of a personal history of metastatic colon cancer, headache and profuse positional vomiting (Figure 2).

## 4. Discussion

This is, to our knowledge, the first study to evaluate the clinical presentation and outcomes of patients presenting with suspected hc-BPPV-cu to the Emergency Department. Here, we have compared such presentation to those that present to a specialist outpatient ENT setting. In a specialist emergency setting, there was a lower disease duration of the hc-BPPV-cu, a smaller number of maneuvers necessary to resolve it, and less evolution to mc-BPPV, together with an earlier diagnosis of a central disorder in patients with central mimics.

Relating to peripheral vestibular BPPV pathology, we find that the current literature [26] describes emerging and controversial syndromes of BPPV, atypical nystagmus according to the canal involved and persistent geotropic positional nystagmus. The study of these peculiarities allows us to better understand the pathomechanisms of BPPV and to differentiate it from central disorders that cause vertigo and nystagmus. In the apogeotropic variant of hc-BPPV, and during the head roll test, the deflection of the heavy cupula in response to the positional changes explain a contralesional head-bending nystagmus (which appears by bending the head forward while sitting) and an ipsilesional lying-down nystagmus (induced by lying supine from the sitting position). Therefore, pseudo-spontaneous nystagmus is described in hc-BPPV [20,21]. It beats toward the affected ear in hc-BPPV-cu, and it is explained by the horizontal canal inclination of 30° backwards from the horizontal plane. It is necessary to differentiate this pseudo-spontaneous nystagmus from the “canalith jam”, which is a mono-positional apogeotropic nystagmus due to canalolithiasis trapped within the ampulla. In patients with a suspected diagnosis of “canalith jam” a central pathology should be suspected. Furthermore, null points of the horizontal nystagmus according to the head position are observed in hc-BPPV-cu, due to the 10–25° lateral inclination of the cupula in the supine position. Such BPPV ocular motor characteristic variants can be challenging, and may lead to a suspicion (incorrectly) of a central pathology, although we did not observe any such atypical nystagmus in our hc-BPPV-cu cohorts.

We observed that hc-BPPV-cu resolved with a single maneuver in 62.5% of cases in the OC and in 75.9% of cases in the ED, with the modified Gufoni maneuver being the most used in both cohorts. This is supported by previous studies such as that of Alvarez de Linera-Alperi et al. [27], where the success rate with a single maneuver was 63% with the modified Gufoni and of 56% with the Zuma e Maia maneuver.

Moreover, our study resulted in less evolution to mc-BPPV of the hc-BPPV-cu in the ED setting. Mc-BPPV usually involves a combination of posterior canal and horizontal canal [24], as in our sample. This is suggested by the presence of both the torsional-vertical nystagmus during Dix-Hallpike maneuver and the horizontal direction-changing nystagmus during supine head roll test [28], as we observed in our patients. The lower rate of evolution of the hc-BPPV-cu to mc-BPPV likely represents the more rapid resolution of BPPV in the ED group in our study.

Central positional nystagmus is an important and consolidated differential diagnosis in BPPV, given the potential for life-threatening or progressive underlying etiologies. Diagnosis of central positional nystagmus remains a challenge due to its similarities to BPPV, and is largely based on atypical features for BPPV rather than its own specific characteristics. Accordingly, our study takes as a relevant reference the largest review of central positional nystagmus to date [29]. Important differentiating factors between BPPV and central positional nystagmus include associated central oculomotor (non-positional) findings (mainly cerebellar), neurological signs, and the absence of improvement with particle displacement maneuvers [23]. In patients with confirmed central pathologies across our two cohorts, the positional nystagmus mimicked a hc-BPPV-cu, so the presence of additional neurological features and failure of resolution of symptoms despite repeated positional maneuvers were central to making the correct diagnosis. Central positional nystagmus is explained by an abnormal integration of the signals related to the semicircular canal, provided by the dysfunction of the cerebellum in any of its parts, which generates an erroneous estimation in the coordination of the head position and of the eye. Thus, the presence of central positional nystagmus is highly predictive of lesions in the posterior fossa, involving a communicating network between the vestibular apparatus (otolith organs and semicircular canals), brainstem vestibular nuclei and midline cerebellar structures within the vermis. On this basis, there may be different clinico-radiological or clinico-pathological central positional nystagmus syndromes [29].

In relation to nystagmus characteristics, spontaneous nystagmus is not a typical feature of central positional nystagmus [29], which could explain its absence in patients suffering from a central disorder in our study. Of note, the oculographic characteristics of central positional nystagmus (e.g., absent latency, failure of nystagmus to habituate or fatigue, direction-changing nystagmus without a change in head position [29]) were not apparent in our patients with central pathologies.

The need to identify ‘atypical’ BPPV features and associated central features is appropriately highlighted in the three patients evaluated at the ED setting and the two patients evaluated at the OC, all of whom were found to a central disorder that mimicked a hc-BPPV-cu. The patient who suffered a bilateral PICA infarction had a strong history of cardiovascular risk factors (arterial hypertension, atherosclerosis and diabetes mellitus), raising our suspicion of a possible ischaemic aetiology [30], together with marked gait ataxia, less typical for BPPV. The degree of vestibulospinal involvement is an increasingly recognized factor for the differential diagnosis of central versus peripheral vestibulopathy, although it has only been confirmed in the context of the acute vestibular syndrome rather than BPPV [31]. However, the presence of ageotropic horizontal nystagmus has also been recognized as a hc-BPPV-cu mimic, as in the case of a unilateral large cerebellar infarct involving the nodulus [32]. In the case of the patient with a dissection of the vertebral artery in our cohort, the positional symptoms were associated with intense neck pain, not consistent with BPPV. In the case of the patient with metastasis in the right cerebellar hemisphere, the personal history of metastatic cancer, together with headache and profuse vomiting, classical symptoms of intracranial hypertension, were strong red flags that prompted further evaluation. In this patient, vomiting may be a reflex of central positional pathology, caused by the metastasis as a space occupying lesion. The cause of vomiting may be related to involvement of cerebellar and brainstem pathways necessary for the integration of vestibular and non-vestibular afferents relating to body position in space [29,33]. In these three cases, a cerebral MRI was performed following the criteria established for the suspicion of central pathology in BPPV [13].

A comprehensive approach to the dizzy patient remains critical even in the setting of an Outpatient Clinic (rather than an acute setting), where careful symptom evolution should be monitored, particularly where the natural history is atypical for that condition (e.g., lack of resolution of nystagmus despite appropriately conducted maneuvers for BPPV [23]). This lack of resolution of hc-BPPV-cu alerted us to diagnose central pathology in all our patients in cohort A. Furthermore, as in the case of the patient with a right hemipontine stroke, the presence of cardiovascular risk factors (arterial hypertension and dyslipidemia) [30], together with otological symptoms atypical for BPPV (hearing loss) should be taken into consideration, even in the presence of positional nystagmus typical for BPPV. Positional nystagmus in hemipontine stroke may be explained by nystagmus mechanisms similar to those in inferior cerebellar peduncle lesions, due to its close anatomical association. The inferior cerebellar peduncle contains cerebellar afferents, including the dorsal spinocerebellar tract and the axons from the vestibular nuclear complex and inferior olivary nucleus. Thus, lesions in these areas affect the integration of proprioceptive, sensory and vestibular signals. This triggers an apogeotropic nystagmus beating to the lesion side (ipsilesional nystagmus) [34]. The patient with cerebellar atrophy was diagnosed in OC, too. This patient showed marked gait ataxia. Concerning this pathology, attempts have already been made to correlate cerebellar atrophy with BPPV, although conclusive results have not been previously obtained [35].

Central pathology was located in cerebellum in three patients in our study. It involved a patient evaluated in OC, who presented cerebellar atrophy, and two patients from the ED, one of them with bilateral PICA infarction, and the other one with metastasis in the cerebellar hemisphere. Central positional nystagmus and vertigo had been previously explained by cerebellar or brainstem lesions around the fourth ventricle which may interrupt the vestibular nucleus-vestibulocerebellar loop as well as the cerebellar nodulus or uvula lesion. Specifically, apogeotropic nystagmus in these lesions may be explained by inhibition of the vestibulo-oculomotor system including the cerebellar flocculonodular lobe or vestibulo-cerebellum, through its connections with the otolith organs and with the vertical semicircular canals. These structures control the otolith-ocular reflexes and otolithic modulation of the semicircular canal-ocular reflexes [36]. Consequently, patients are often first misdiagnosed with hc-BPPV-cu due to apogeotropic nystagmus and positional vertigo [37], as reflected in our sample.

It is important to note however that hc-BPPV-cu may coexist with other independent central structural or functional pathologies. In such cases, one must be mindful to correctly diagnose and treat both pathologies.

We acknowledge some limitations to this study, namely the small sample size and the inability to standardize the patient follow up such that assessments did not necessarily take place at the same time intervals for all patients. Therefore, we recommend future research should focus on the specific central symptoms that mimic hc-BPPV-cu most frequently, the most frequent positional nystagmus characteristics that may be associated with central nystagmus, and investigating the underlying mechanism of central nystagmus. Larger studies may help corroborate or complement our findings.

## 5. Conclusions

In summary, a comprehensive approach to the patient with cupulolithiasis of the horizontal semicircular canal presenting to the Emergency Department requires careful evaluation of associated features, identification of possible red flags for a central pathology, longer term follow up of treatment outcome, and early MRI imaging where atypical features are identified. Thus, this approach allows us to be both more effective in hc-BPPV-cu treatment and to rule out central pathologies that mimic it earlier, given the importance of an early diagnosis to minimise potential morbidity and mortality.

## Figures and Tables

**Figure 1 brainsci-13-00562-f001:**
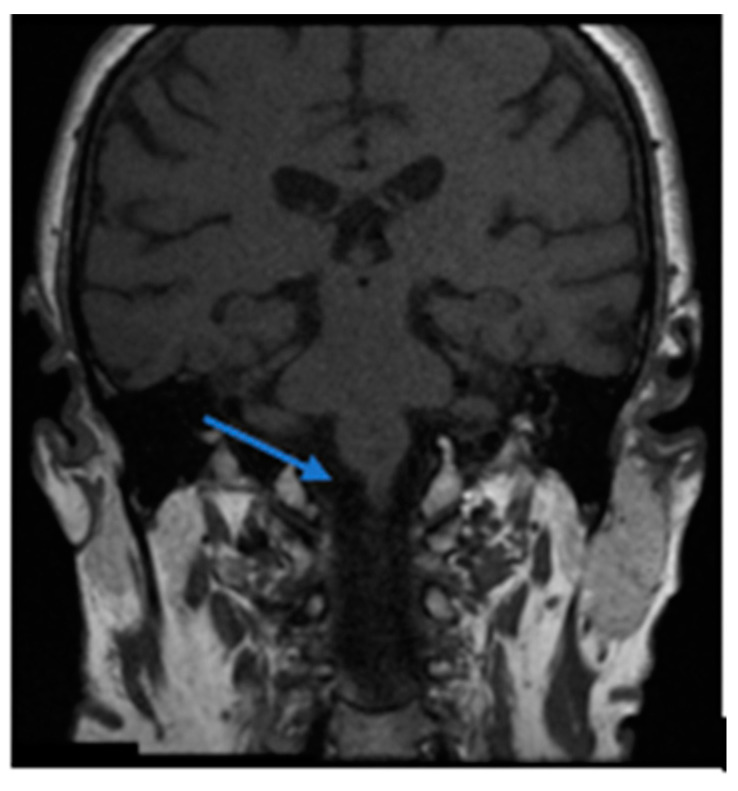
Right hemipontine ischemia (blue arrow). This patient associated cardiovascular risk factors (arterial hypertension and dyslipidemia) together with right-sided hearing loss and unsteadiness.

**Figure 2 brainsci-13-00562-f002:**
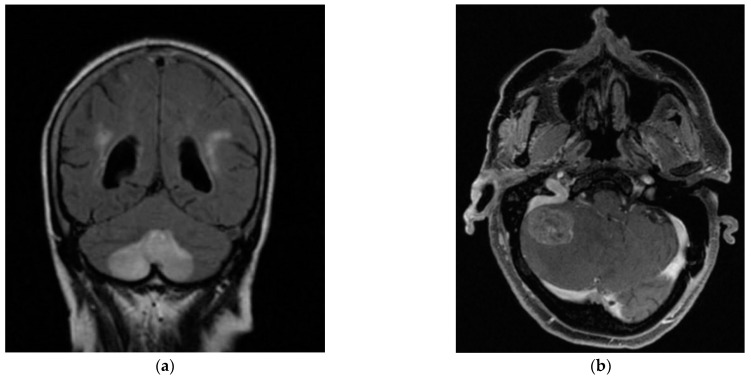
Bilateral posterior inferior cerebellar artery (PICA) infarction (**a**), in a patient with cardiovascular risk factors (arterial hypertension, atherosclerosis and diabetes mellitus). Metastasis in the right cerebellar hemisphere (**b**) associated with a personal history of metastatic colon cancer and headache and profuse vomiting.

**Figure 3 brainsci-13-00562-f003:**
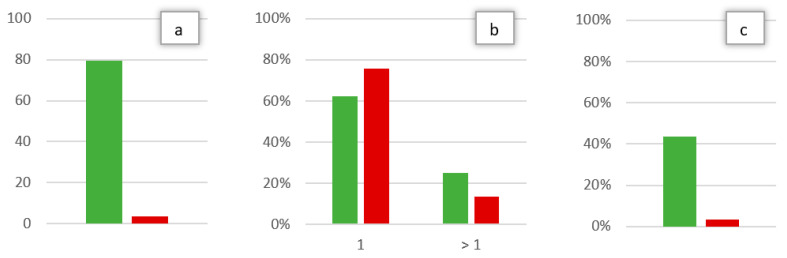
Values with a statistically significant difference between cohort A (OC, in green) and B (ED, in red): Time from symptom onset to first evaluation in days (**a**), percentage of maneuvers necessary for the resolution of the hc-BPPV-cu (one or more than one) (**b**), percentage of patients with mc-BPPV at the time of hc-BPPV-cu diagnosis (**c**).

**Table 1 brainsci-13-00562-t001:** Main demographic and clinic details. * denotes statistical significance at *p* < 0.05.

		Cohort A (OC) *n* = 16			Cohort B (ED) *n* = 29			*p*-Value
Gender (Male/Female) (%)		37.5%/62.5%			55.2%/44.8%			0.353
Age (years)		70.5 (SD 12.65)			67.07 (SD 14.09)			0.355
Days from vertiginous symptoms onset to first evaluation		79.7			3.6			0.002 *
Prior history of BPPV		31.3%	pc-BPPV	100%	27.6%	pc-BPPV	6.9%	0.528
						hc-BPPV	10.3%	
mc-BPPV		43.8%			3.4%			0.002 *
Number of maneuvers for resolution (%)	1	62.5%			75.9%			0.004 *
	>1	25.1%			13.7%			0.003 *
	Unresolved	12.5%			10.3%			0.512

## Data Availability

The data presented in this study are available on request from the corresponding author. The data are not publicly available due to privacy.

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
