# Peer review of "Early Diagnosis of Central Disorders Mimicking Horizontal Canal Cupulolithiasis"

_brainsci, 2023, doi:10.3390/brainsci13040562_

Round 1

Reviewer 1 Report

The paper deals with important clinical problem – differential diagnosis of positional nystagmus. The timely recognition of central versus labyrinthine aetiology is crucial especially in emergent setting.

The authors have given thorough overview of this problematic and were able to summarize number of cases of this not very frequent condition – horizontal canal cupulolithiasis. This disease is always a challenge and risk to miss a central posterior fossa pathology is high.

To increase awareness of this peculiar syndrome is important for clinical practice in neurology, ENT and emergency medicine.

This is a retrospective study comparing  results from ENT versus Emergency departments. The main difference was much longer disease duration in ENT department, but this is not a surprise – the chronic cases probably do not attend emergency for long standing problems.

More interesting fact is, that one single  liberatory manoeuvre healed most of the patients – even the chronic ones in ENT dept. This is worth to more detailed explanation in Discussion, what is the comparison eg. to other studiesThe paper is well written, with no mistyping.

row 78 :   bow and lean maneuver – this is not a name of an author!

when the laterality of the affected canal is in doubt, the Bow and Lean maneuver

The plan for statistical evaluation is extensive, but little from it is presented in the paper – no results of Chi square, correlation etc.  – see rows 82-90!

From methodology it is event not clear, what other variables were analysed, aside from gender, age, diagnosis and group assignment…

It would be useful to give more detailed description of the five central cases and specifically pinpoint to red flag symptoms and signs, which were critical for correct differential diagnosis!

In the case of cerebellar metastasis it should be stated, that this is classical case of intracranial hypertension - “eg. severe vomiting and other symptoms of intracranial hypertension.”

I will also suggest to shorten the statistical plan and mention only technique presented in the text.

The paper should be published after minor changes mentioned above.

Reviewer 2 Report

Review: Early diagnosis of central disorders….

 The authors analyzed data in regard to horizontal semicircular canal BPPV comparing outpatient (OC) and emergency room (ER) diagnostics/therapies with aspects of differential diagnosis on non-BPPV etiologies.

 While the data are interesting, the authors do not do due diligence to the data set. The authors seem to be ambivalent about the direction their study and analysis is going, i.e., do they compare outcomes between OC and ER environments. The differential diagnostics regarding BPPV and “central disorders” come more like as an afterthought. Therefor, the authors need to be clear what the impetus of their study should be. Also, they need to discuss the discrepancies between the findings in the two patient cohorts. To me, as an aficionado, it is clear, but the authors need to address the broad readership of the journal.

  Aside from this aspect, there are some minor issues to be take care of.

 Please, specify mc-BPPV. Which canals were involved?

 Instead of “particle replacement” use “particle displacement” throughout.

l. 31: “..produced by inadequate mechanical….”

l. 53: The first sentence of the "Materials." does not make sense. Maybe they should start out: “We compared two patient cohorts that presented with BPPV to either the outpatient clinic (OC: Group A) or the ENT Emergency Room (ER; Group B) with regard to outcomes (diagnostics and therapies) of Salamanca University. Our analysis also took into consideration differential diagnosis between true BPPV and central disorders mimicking BPPV.”

 L. 181: “..were associated with intense….”

Fig. 2. Why not use “a”, “b” in the legend (instead of “left”, “right”

 Fig. 3. Use the actual numbers instead of %.

Reviewer 3 Report

Hc-BPPV is affecting about 25% of elderly people that can with a single maneuver be resolved. The remaing  required more maneuvers except that were not resolved that correlates with cardiovascular diseases. Limitations are the smallest sample size...

Line 42;  suggests citations: Türk, Bilge, et al. Ear, Nose & Throat Journal 100.7 (2021): 532-535; Young, Allison S., et al." Neurology 92.24 (2019): e2743-e2753.

Line 45ff cite additional work: Fu, Wei, et al. Auris Nasus Larynx 47.1 (2020): 48-54.; Koju, Geeta, et al. Current Medical Science 42.3 (2022): 613-619.
